# Association of psychological distress, quality of life and costs with carpal tunnel syndrome severity: a cross-sectional analysis of the PALMS cohort

Christina Jerosch-Herold,[1] Julie Houghton,[1] Julian Blake,[2,3] Anum Shaikh,[4] Edward CF Wilson,[4] Lee Shepstone[5]

For numbered affiliations see end of article.

**Correspondence to**
Professor Christina Jerosch-Herold;
c.jerosch-herold@uea.ac.uk

## ABSTRACT

**Objectives** The Prediciting factors for response to treatment in carpal tunnel syndrome (PALMS) study is designed to identify prognostic factors for outcome from corticosteroid injection and surgical decompression for carpal tunnel syndrome (CTS) and predictors of cost over 2 years. The aim of this paper is to explore the cross-sectional association of baseline
patient-reported and clinical severity with anxiety, depression, health-related quality of life and costs of CTS in patients referred to secondary care.

**Methods** Prospective, multicentre cohort study initiated in 2013. We collected baseline data on patient-reported symptom severity (CTS-6), psychological status (Hospital Anxiety and Depression Scale), hand function (Michigan Hand Questionnaire) comorbidities, EQ-5D-3L (3-level version of EuroQol-5 dimension) and sociodemographic variables. Nerve conduction tests classified patients into five severity grades (mild to very severe). Data were analysed using a general linear model.

**Results** 753 patients with CTS provided complete baseline data. Multivariable linear regression adjusting for age, sex, ethnicity, duration of CTS, smoking status, alcohol consumption, employment status, body mass index and comorbidities showed a highly statistically significant relationship between CTS-6 and anxiety, depression and the EQ-5D (p<0.0001 in each case). Likewise, a significant relationship was observed between electrodiagnostic severity and anxiety (p=0.027) but not with depression (p=0.986) or the EQ-5D (p=0.257). National Health Service (NHS) and societal costs in the 3 months prior to enrolment were significantly associated with self-reported severity (p<0.0001) but not with electrodiagnostic severity.

**Conclusions** Patient-reported symptom severity in CTS is significantly and positively associated with anxiety, depression, health-related quality of life, and NHS and societal costs even when adjusting for age, gender, body mass index, comorbidities, smoking, drinking and occupational status. In contrast, there is little or no evidence of any relationship with objectively derived CTS severity. Future research is needed to understand the impact of approaches and treatments that address psychosocial stressors as well as biomedical factors on relief of symptoms from carpal tunnel syndrome.

## INTRODUCTION

Carpal tunnel syndrome (CTS) is the most common nerve entrapment syndrome; an

estimated 1 in 10 people are likely to develop symptoms at some stage in their lives.[1] CTS is characterised by pins and needles, pain and numbness, often affecting both hands. Symptoms can range from intermittent to constant and from very mild to very severe, potentially interfering significantly with daily activities.[2] Approximately one-third to two-thirds of patients with CTS go on to have surgery.[3–5] Such variations in surgical rates may be due to different follow-up periods in individual studies as well as differences in criteria for and access to surgical decompression.

### Strengths and limitations of this study

► Large multicentre prospective cohort study of carpal tunnel syndrome (CTS) (n>700).
► We used a multivariable model to test the independence of associations with a range of prespecified demographic and clinical variables.
► All patients had a diagnosis of CTS made through the combination of clinical history, signs and symptoms with objective electrodiagnostic testing.
► Cross-sectional analysis limited to baseline data only with longitudinal (2 years) follow-up still in progress.
► Cost data are limited to a 3-month horizon prior to enrolment.

Up to 54000 carpal tunnel decompressions were carried out in England in 2014–2015 (Hospital Episode Statistics) at a cost of £46 million to the National Health Service (NHS) (based on an NHS tariff of £864) and are predicted to increase twofold by 2030.[6] Clinical Commissioning Groups (CCG) in England have introduced guidelines on funding for procedures deemed to be of 'limited clinical value', recommending carpal tunnel surgery only for those with '*moderate to severe CTS*' or who have '*not responded to conservative measures*' (ie, corticosteroid injections

and splints). However, it is often unclear whether severity refers to pathophysiology or symptoms or a combination of both. A recent review of policies from 175 CCGs in England highlighted that these were highly variable in terms of criteria for defining severity, the necessity for nerve conduction studies (NCS) and duration of non-operative treatment before referral for surgery is allowed.[7] Controversy continues over the best management of CTS at different stages. More recent research on CTS has highlighted the role of psychological factors such as anxiety, depression and patient's health beliefs in how patients perceive their symptoms, their impact on function and the outcome from treatment.[8–10] No previous studies have explored the association of disease and symptom severity with health-related quality of life (HRQoL) or costs from a personal and societal perspective and which may also play an important role in treatment planning.

The objectives of this cross-sectional analysis were (1) to explore the association of self-reported symptom severity and electrodiagnostic severity with anxiety, depression, hand function and HRQoL in patients referred for diagnosis and treatment of CTS; and (2) to describe NHS and societal costs of CTS in the 3 months preceding referral for nerve conduction testing and treatment and their association with carpal tunnel severity.

## METHODS

### Design and study population

A prospective longitudinal cohort study (PALMS study) was initiated in 2013 to develop a multivariable prognostic model of predictive factors for outcome after steroid injection, outcome after surgery and costs in CTS. The full study protocol has been described elsewhere.[11] Recruitment took place at five secondary care sites across five NHS trusts in England where patients had been referred by primary care for nerve conduction tests and prior to any treatment. Patients were diagnosed with CTS by a neurophysiologist or hand surgeon based on signs and symptoms, clinical history and objective measurable pathophysiology (NCS). Eligible patients were invited to participate by returning completed screening questionnaires and signed consent forms. Patients were included if they had CTS in at least one hand confirmed by NCS (grade ≥1) and were aged 18 years or over. NCS grading was done according to Bland's criteria[12] (for details, see online supplementary file 1). Patients with concomitant diseases such as diabetes or hypothyroidism were included. Exclusion criteria were previous surgery in the affected hand in the last 12 months, pregnancy or up to 12 months post partum, serious comorbidities, other limb mononeuropathies, sensory or motor disturbances secondary to stroke, multiple sclerosis or nerve injury and inability to speak or write English.

For the present study, data collected at baseline were used. Self-reported symptom severity and NCS for only one hand per patient were used. In patients with bilateral CTS, the 'index hand' was defined as either the worst hand or the dominant hand where self-reported symptom severity on the CTS-6 was the same bilaterally.

### Data collection

A baseline questionnaire was developed comprising the following standardised and non-standardised questionnaires:

► Symptom severity was assessed with the six-item shortened form of the Boston Carpal Tunnel Questionnaire,[13] the CTS-6[14] for each hand. The scores range from 1 (no symptoms) to 5 (very severe symptoms). A mean score of 1 indicates no symptoms (tingling, numbness and pain during the past 2 weeks).

► Hand functional status was assessed with three subscales of the Michigan Hand Questionnaire (MHQ)[15 16]: overall hand function for worst hand, unilateral and bilateral activities of daily living and work performance. Each subscale is converted into a percentage with higher values denoting greater disability.

► Psychological status was assessed using the Hospital Anxiety and Depression Scale (HADS).[17] It is made up of 14 items, 7 relating to anxiety and 7 relating to depression. Responses are scored from 0 to 3 giving a possible score range of 0–21 for each subscale. Scores of 8–10 identify mild cases, 11–15 moderate cases and 16 or more severe cases.[17]

► HRQoL was assessed by EQ-5D-3L (3-level version of EuroQol-5 dimension).[18]

► Comorbidities were collected by the Self-Administered Comorbidity Questionnaire (SACQ).[19] The total score ranges from 0 to 36 which is based on 12 listed conditions for which the patients indicate if they have it (1 point), receive treatment for it (2 points) and whether it is activity limiting (3 points).

► Healthcare resource use in the 3 months prior to enrolment. This included questions about treatments received (including alternative therapies), medications (prescription and over the counter), healthcare contacts (NHS primary, secondary and tertiary, and private), patient-reported days off work due to CTS and assistance with activities of daily living due to CTS.

Other clinical and demographic variables collected at baseline were age, sex, duration of symptoms, height and weight, work status and type, smoking status, alcohol units consumed per week, ethnicity and household income. Additionally, the full NCS reports of enrolled patients were obtained from the participating sites and were graded by the first author using Bland's criteria[12] from 1 (mild) to 6 (extremely severe) (see online supplementary file 1).

Patients were given the option of completing the study questionnaires online via a personalised link to a password-protected web-based data entry system maintained by the Norwich Clinical Trials Unit or via paper-based questionnaires sent by mail.

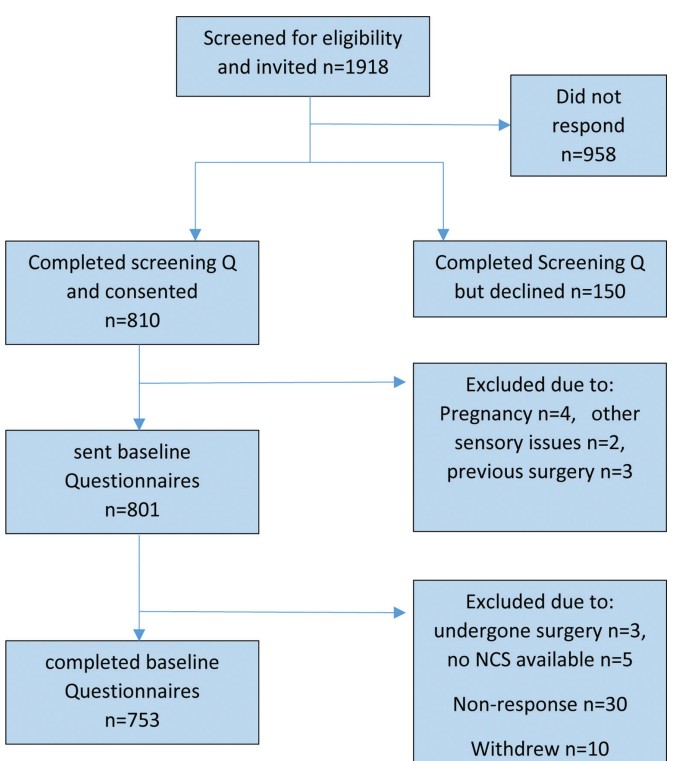

**Figure 1** STROBE (Strengthening the Reporting of Observational Studies in Epidemiology) flow chart.[31] NCS, nerve conduction studies.

## Statistical analysis

A generalised linear model (GLM) with a normally distributed error term was used to model the HADS anxiety score, HADS depression score, MHQ scores and EQ-5D utility index. EQ-5D responses were converted to utilities using UK-specific preference weights.[20] The model included gender, age, ethnicity, employment status, smoking status, units of alcohol per week, body mass index (BMI), duration of CTS and comorbidity score as explanatory variables before adding either CTS-6 symptom score or NCS score. Parameter estimates for the effect of CTS-6 and NCS score are provided with 95% CIs and statistical significance set at the conventional 5% level.

Resource use data were converted to cost per patient using standard UK unit cost sources (Personal Social Services Research Unit, British National Formulary and Office for National Statistics), the relevant published literature and consultation with experts. Costs were then modelled using a GLM with log-link and gamma distributed errors. The base year for the costs was 2015/2016 and the analysis was conducted from the perspectives of the NHS and society (defined as the sum of NHS and patient out-of-pocket costs and morbidity-related lost productivity).

## RESULTS

Recruitment took place over 30 months between July 2013 and December 2015. A total of 1918 patients with CTS were identified and invited to participate of which a total of 753 patients met all eligibility criteria and returned

| Table 1 | Demographics |
| --- | --- |
| | **Number (%)** |
| **Sex** | |
| Male | 260 (35%) |
| Female | 493 (65%) |
| **Age (years)** | |
| Mean (SD) | 60.3 (12.7) |
| **Ethnicity** | |
| White | 731 (97%) |
| Other | 20 (3%) |
| Missing | 2 |
| **BMI** | |
| Mean (SD) | 28.7 (5.8) |
| **Smoking status** | |
| Current | 76 (10%) |
| Ex-smoker | 299 (40%) |
| Non-smoker | 375 (50%) |
| Missing | 3 |
| **Alcohol intake (units)** | |
| None | 251 (34%) |
| 1–4 | 271 (37%) |
| 5–10 | 114 (15%) |
| 11–20 | 81 (11%) |
| >21 | 25 (3%) |
| Missing | 11 |
| **Working status** | |
| Working | 296 (39%) |
| Self-employed | 88 (11.7%) |
| Not working | 457 (61%) |
| Retired | 322 (43%) |
| Unemployed | 14 (1.9%) |
| Long-term sick | 36 (4.8%) |
| Carer, volunteer or student | 85 (11.75%) |
| **Income (per annum)** | |
| <£15K | 170 (23%) |
| £15K–£21.5K | 103 (14%) |
| £21.5–£35K | 148 (20%) |
| £35K–£50K | 81 (11%) |
| >£50K | 49 (7%) |
| Missing | 5 |

BMI, body mass index.

full baseline questionnaires (see STROBE (Strengthening the Reporting of Observational Studies in Epidemiology) diagram in figure 1).

The sociodemographic and clinical variables of the cohort are summarised in tables 1 and 2. Bilateral CTS affected 69% of patients and 75% of participants had their symptoms for at least 6 months or longer. Using their

| Table 2 Clinical characteristics | |
|---|---|
| | **Number (%)** |
| Duration of CTS | |
| <3 months | 42 (6%) |
| 3–6 months | 140 (19%) |
| 6–12 months | 181 (24%) |
| 12–18 months | 98 (13%) |
| >18 months | 292 (39%) |
| Side of CTS | |
| Left | 89 (12%) |
| Right | 148 (20%) |
| Bilateral | 516 (69%) |
| Dominant hand affected | |
| Yes | 669 (89%) |
| No | 84 (11%) |
| Worst side | |
| Left | 264 (35%) |
| Right | 489 (65%) |
| Worst Side NCS Grade | |
| 1 | 150 (20%) |
| 2 | 79 (11%) |
| 3 | 202 (27%) |
| 4 | 137 (18%) |
| 5 | 159 (21%) |
| 6 | 24 (3%) |
| Missing | 2 |
| CTS-6 (1–5) | |
| Mean (SD) | 2.89 (0.85) |
| MHQ overall function (0–100) | |
| Mean (SD) | 53.4 (22.6) |
| MHQ unilateral activities (0–100) | |
| Mean (SD) | 66.6 (28.6) |
| MHQ bilateral activities (0–100) | |
| Mean (SD) | 69.4 (25.5) |
| MHQ work (0–100) | |
| Mean (SD) | 64.7 (25.6) |
| HADS Anxiety | |
| Mean (SD) | 6.18 (4.52) |
| Normal (1–7) | 497 (66%) |
| Mild (8–11) | 123 (16%) |
| Moderate (11–14) | 90 (12%) |
| Severe (15–21) | 42 (6%) |
| Missing | 1 |
| HADS Depression | |
| Mean (SD) | 4.48 (3.84) |
| Normal (1–7) | 602 (80%) |
| Mild (8-10) | 88 (12%) |

Continued

| Table 2 Continued | |
|---|---|
| | **Number (%)** |
| Moderate (11–14) | 47 (6%) |
| Severe (15–21) | 15 (2%) |
| Missing | 1 |
| Overall Co-morbidity Score (0–36) | |
| Mean (SD) | 5.22 (4.17) |
| EQ-5D-3L VAS | |
| Mean (SD) | 73.54 (18.2) |

CTS, carpal tunnel syndrome; EQ-5D-3L, 3-level version of EuroQol-5 dimension; HADS, Hospital Anxiety and Depression Scale; MHQ, Michigan Hand Questionnaire; NCS, nerve conduction studies; VAS, visual analogue scale.

worst hand as the index hand, only 18% had received a steroid injection and mean self-reported symptom severity (CTS-6) was 2.89 points. Based on nerve conduction tests, 69% of patients had at least moderate CTS (grades ≥3), with 24% in the severe and very severe categories (grade 5 or 6) (see table 2).

The mean CTS-6 scores increased steadily with NCS grade up to grade 6 (which contained relatively few individuals). The mean (and SD) by grade were as follows: grade 1, 2.69 (0.86); grade 2, 2.79 (0.85); grade 3, 2.87 (0.91); grade 4, 3.00 (0.79); grade 5, 3.06 (0.79); grade 6, 2.89 (0.78).

The HADS anxiety and HADS depression scores were within normal ranges (scores 0–7) in 66% and 80% of participants, respectively. Self-reported comorbidities are summarised in table 3. The mean SACQ score was 5.22. The three most common comorbidities were low back pain

| Table 3 Self-reported comorbidities | | | |
|---|---|---|---|
| Comorbidity | Has problem (%) | Receives treatment (%) | Limits activity (%) |
| Back pain | 342 (45) | 142 (19) | 226 (30) |
| Osteoarthritis | 253 (34) | 129 (17) | 185 (25) |
| High blood pressure | 235 (31) | 226 (30) | 22 (3) |
| Depression | 148 (20) | 117 (16) | 63 (8) |
| Diabetes | 77 (10) | 63 (8) | 11 (1) |
| Rheumatoid arthritis | 70 (9) | 40 (5) | 51 (7) |
| Lung disease | 29 (4) | 23 (3) | 18 (2) |
| Cancer | 18 (2) | 13 (2) | 5 (<1) |
| Kidney disease | 14 (2) | 5 (<1) | 3 (<1) |
| Liver disease | 4 (<1) | 1 (<1) | 0 |
| Ulcer or stomach disease | 37 (5) | 32 (4) | 7 (1) |
| Other condition 1 | 372 (49) | 280 (37) | 150 (20) |
| Other condition 2 | 140 (19) | 95 (13) | 60 (8) |
| Other condition 3 | 48 (6) | 30 (4) | 20 (3) |

(LBP), osteoarthritis (OA) and high blood pressure, with LBP and OA also reported as the most activity limiting. Mean visual analogue scale score on the EQ-5D-3L was 73.5 (SD=18.2), and the mean health utility index was 0.66. Over 85% of respondents indicated having at least some pain and 31% of the cohort indicated having both pain and feeling anxious or depressed.

Using a multivariable regression model including age, sex, ethnicity, duration of CTS, smoking status, alcohol consumption, employment status, BMI and comorbidity score as potential confounding variables, a highly statistically significant association was found between self-reported symptom severity (using CTS-6 score) and anxiety, depression and HRQoL (p<0.0001 in each case). Table 4A gives the parameter estimates for psychological outcome and quality of life by self-reported and objectively graded severity. For each 1 point increase in score on the CTS-6 there was an estimated 1.11 points increase in anxiety score, 1.43 points increase in depression score and a 10% mean decrease in health utility. Patients with the highest symptom severity (4–5 points on CTS-6) had mean anxiety scores of 9.62 (SD=5.48) indicating at least mild anxiety. There was a marked lower health utility for those with the highest symptom severity (scores 4–5 on CTS-6) with a mean health utility index of 0.43 compared with 0.77 in the mildest group (scores 1–2 on CTS-6). When including the same independent variables in the model, objective severity grading based on NCS was significantly associated with anxiety (p=0.027), but not with depression or HRQoL. The association between NCS grade and anxiety, however, was negative with higher anxiety scores observed in severity grades 1 and 2 and the lowest mean anxiety scores observed in the worst severity group (mean 3.42). For every one grade increase in neurophysiologically assessed severity there was an estimated 0.26 point decrease in the HADS anxiety score. Using the same multivariable regression model, a highly significant independent association was also found between symptom severity and hand function subscales. For every 1 point increase in symptom severity, there was a marked decrease in hand function ranging from 13.4 to 17.8 points on the MHQ (see table 4B for parameter estimates by subscales). Electrodiagnostic severity was also significantly associated with overall hand function and unilateral and bilateral activities of daily living but not with the work subscale. The magnitude of decline in hand function was less marked when using NCS grade in the model.

The mean cost of NHS service use per patient in the 3 months prior to baseline assessment was £447 (SD=£274), with a societal cost (NHS plus patient out of pocket and lost productivity) of £636 (SD=£694). NHS costs comprised predominantly hospital visits including NCS (£362), general practitioner (GP) consultations (£58), prescription medications (£14) and other contacts and treatments (£13), which accounted for 70% of societal costs. Lost productivity (£101) accounted for 15% and out-of-pocket costs (£56 for lost wages due to GP/hospital visits, £7 for travel to the hospital, £4 for hand

splints, and £18 for other contacts and treatments for the remaining 15%). Although those individuals over the age of 40 generally have higher NHS and lower societal costs compared with 18–40 year-olds (reference category), age is not a significant predictor of costs under either perspective. For example, NHS costs are 15% higher on average for those individuals in the 81+ age group, when compared with 18–40 year-olds and accounting for NCS grade, but this, however, is not a significant determinant of costs (p=0.327). NHS and societal costs, however, significantly increase with self-reported severity of CTS (p<0.01 and p<0.0001), but not NCS grade (p=0.269 and p=0.590). For every point increase in the self-reported severity, NHS and societal costs increase by 8% and 18%, respectively (see table 5).

## DISCUSSION
### Principal findings
We found that at the point of referral for NCS and prior to any treatment, greater symptom severity was associated with greater psychological distress, poorer hand function and lower quality of life but not with worse electrodiagnostic abnormalities. Even after adjusting for known and potential confounders such as comorbidities, age, gender and BMI, we observed that every 1 point increase in symptom severity score was associated with an estimated 1.11 points increase in mean anxiety score, a 1.43 points increase in depression score and a 0.10 decrease in health utility. Adjusting for the same independent variables, objectively graded severity from NCS, however, showed a significant negative association with anxiety and no significant association with depression or health utility. When modelling the outcome on self-reported hand function (MHQ), both symptom severity and NCS showed a significant association with overall hand function and activities of daily living, though the decrease in hand function was much more marked when using symptom severity (CTS-6). More than 40% of the variation in hand function was explained by the model when using CTS-6 score and less than 20% when using NCS grade in the model.

Average NHS and societal costs per patient were £447 and £636, respectively. There was no significant association between costs, age and objectively graded severity. We found, however, for every point increase in subjective severity, an 8% and 18% relative increase in NHS and societal costs, respectively.

Several other studies have explored the association of pain severity with a range of psychological variables including anxiety, depression, pain catastrophising and coping.[9 10 21] These studies all conclude that illness behaviour is a stronger predictor of pain severity than objective measures of disease severity (NCS). In our study, we did not measure pain intensity, although the CTS-6 does contain three items relating to pain (daytime pain,

**Table 4A** Relationship between CTS severity and anxiety, depression and quality of life

| n | | HADS anxiety Mean (SD) | Estimate (95% CI)* | HADS depression Mean (SD) | Estimate (95% CI)* | EQ-5D Mean (SD) | Estimate (95% CI)* |
|---|---|---|---|---|---|---|---|
| **CTS-6 score** | | | | | | | |
| 1.00 | 7 | 3.14 (2.61) | | 1.57 (1.81) | | 0.78 (0.32) | |
| 1.01–2.00 | 134 | 5.04 (4.07) | | 2.90 (3.12) | | 0.77 (0.20) | |
| 2.01–3.00 | 308 | 5.50 (4.05) | | 3.86 (3.31) | | 0.72 (0.20) | |
| 3.01–4.00 | 241 | 6.87 (4.55) | | 5.42 (3.99) | | 0.60 (0.28) | |
| 4.01–5.00 | 63 | 9.62 (5.48) | 1.11 (0.73 to 1.49) | 7.68 (4.44) | 1.43 (1.13 to 1.73) | 0.43 (0.34) | −0.10 (−0.12 to −0.08) |
| | | | p<0.0001 | | p<0.0001 | | p<0.0001 |
| | | | $R^2$=17.7% | | $R^2$=26.1% | | $R^2$=37.4% |
| **NCS grade** | | | | | | | |
| 1 | 150 | 6.78 (4.79) | | 4.64 (4.10) | | 0.66 (0.28) | |
| 2 | 79 | 7.13 (4.78) | | 4.78 (3.73) | | 0.68 (0.25) | |
| 3 | 202 | 6.44 (4.36) | | 4.33 (3.77) | | 0.68 (0.27) | |
| 4 | 137 | 6.24 (4.54) | | 4.80 (3.92) | | 0.68 (0.26) | |
| 5 | 159 | 5.18 (4.24) | | 4.20 (3.80) | | 0.66 (0.24) | |
| 6 | 24 | 3.42 (3.09) | −0.26 (−0.49 to −0.03) | 4.08 (3.13) | −0.00 (−0.20 to 0.20) | 0.61 (0.26) | −0.01 (−0.03 to 0.01) |
| | | | p=0.029 | | p=0.983 | | p=0.214 |
| | | | $R^2$=14.3% | | $R^2$=17.1% | | $R^2$=28.4% |

*Parameter estimate (95% CI) from a linear model including gender, age, ethnicity, employment status, smoking status, units of alcohol per week, body mass index, duration of CTS and comorbidity score.
CTS, carpal tunnel syndrome; EQ-5D, EuroQol-5 dimension; HADS, Hospital Anxiety and Depression Scale; NCS, nerve conduction studies.

**Table 4B**  Relationship between CTS severity and MHQ subscales

| | n | Overall function Mean (SD) | Estimate (95% CI)* | ADL unilateral Mean (SD) | Estimate (95% CI)* | ADL bilateral Mean (SD) | Estimate (95% CI)* | Work Mean (SD) | Estimate (95% CI)* |
|---|---|---|---|---|---|---|---|---|---|
| **CTS-6 score** | | | | | | | | | |
| 1.00 | 7 | 92.1 (10.7) | | 91.4 (16.5) | | 88.8 (11.4) | | 92.9 (12.9) | |
| 1.01–2.00 | 134 | 68.5 (20.3) | | 82.9 (22.0) | | 82.9 (18.9) | | 78.7 (22.3) | |
| 2.01–3.00 | 308 | 58.3 (19.1) | | 74.5 (23.1) | | 76.2 (20.7) | | 70.7 (22.8) | |
| 3.01–4.00 | 241 | 44.9 (18.9) | | 56.8 (26.3) | | 60.7 (24.9) | | 55.0 (23.2) | |
| 4.01–5.00 | 63 | 25.8 (16.2) | −14.7 (−16.4 to −13.1) | 28.2 (26.4) | −17.8 (−19.9 to −15.8) | 38.3 (26.2) | −14.6 (−16.4 to −12.8) | 39.6 (22.2) | −13.4 (−15.4 to −11.5) |
| | | | p<0.0001 | | p<0.0001 | | p<0.0001 | | p<0.0001 |
| | | | R²=41.1% | | R²=40.7% | | R²=40.8% | | R²=33.3% |
| **NCS grade** | | | | | | | | | |
| 1 | 150 | 58.9 (23.9) | | 71.0 (27.9) | | 72.7 (24.4) | | 63.9 (27.4) | |
| 2 | 79 | 53.5 (22.9) | | 71.5 (27.7) | | 74.1 (25.2) | | 67.5 (26.2) | |
| 3 | 202 | 54.6 (22.8) | | 70.5 (27.7) | | 72.6 (24.6) | | 65.8 (26.0) | |
| 4 | 137 | 53.1 (20.5) | | 67.0 (26.3) | | 68.2 (24.3) | | 63.4 (23.9) | |
| 5 | 159 | 48.8 (21.2) | | 57.8 (30.0) | | 63.5 (27.3) | | 64.4 (24.4) | |
| 6 | 24 | 43.3 (24.9) | −3.16 (−3.56 to −1.22) | 47.9 (29.8) | −3.87 (−5.32 to −2.42) | 52.5 (26.0) | −3.16 (−4.42 to −1.90) | 61.9 (23.6) | −0.86 (−2.18 to 0.47) |
| | | | p<0.0001 | | p<0.0001 | | p<0.0001 | | p=0.197 |
| | | | R²=15.5% | | R²=18.7% | | R²=21.9% | | R²=15.7% |

*Parameter estimate (95% CI) from a linear model including gender, age, ethnicity, employment status, smoking status, units of alcohol per week, body mass index, duration of CTS and comorbidity score.
ADL, activities of daily living; CTS, carpal tunnel syndrome; MHQ, Michigan Hand Questionnaire; NCS, nerve conduction studies.

**Table 5** Relationship between CTS severity, age and cost

| | n | NHS cost (SD) | Exp(b) (95% CI)* | Societal cost (SD) | Exp (b) (95% CI)* |
|---|---|---|---|---|---|
| **CTS-6** | | | | | |
| 1.00 | 7 | £512.83 (£405.61) | | £726.02 (£776.80) | |
| 1.01–2.00 | 134 | £434.04 (£212.53) | | £559.37 (£430.53) | |
| 2.01–3.00 | 308 | £422.57 (£220.30) | | £554.73 (£492.25) | |
| 3.01–4.00 | 241 | £466.50 (£348.74) | | £686.11 (£814.25) | |
| 4.01–5.00 | 63 | £511.52 (£273.65) | 1.08 (1.03 to 1.12) p<0.01 | £984.19 (£1197.94) | 1.18 (1.10 to 1.27) p<0.0001 |
| **NCS grade** | | | | | |
| 1 | 150 | £483.36 (£456.12) | | £629.69 (£608.90) | |
| 2 | 79 | £459.44 (£243.93) | | £592.70 (£381.55) | |
| 3 | 202 | £438.50 (£213.31) | | £657.58 (£755.06) | |
| 4 | 137 | £414.14 (£153.80) | | £578.23 (£597.95) | |
| 5 | 159 | £439.38 (£203.02) | | £676.24 (£892.99) | |
| 6 | 24 | £467.71 (£237.77) | 0.98 (0.96 to 1.01) p=0.269 | £583.77 (£294.98) | 1.01 (0.96 to 1.07) p=0.590 |
| **Age and CTS-6** | | | | | |
| Ages 18–40 | 43 | £433.41 (£178.60) | | £758.67 (£1099.72) | |
| Ages 41–60 | 344 | £444.53 (£226.94) | 1.04 (0.88 to 1.22) p=0.679 | £613.03 (£533.45) | 0.83 (0.63 to 1.10) p=0.196 |
| Ages 61–80 | 322 | £451.96 (£336.62) | 1.13 (0.94 to 1.36) p=0.190 | £631.64 (£731.18) | 0.90 (0.66 to 1.22) p=0.486 |
| Ages 81+ | 44 | £442.46 (£146.20) | 1.16 (0.91 to 1.47) p=0.224 | £712.66 (£973.69) | 1.09 (0.73 to 1.62) p=0.690 |
| **Age and NCS grade** | | | | | |
| Ages 18–40 | 43 | £433.41 (£178.60) | | £758.67 (£1099.72) | |
| Ages 41–60 | 343 | £443.81 (£226.88) | 1.02 (0.87 to 1.21) p=0.774 | £608.42 (£527.83) | 0.81 (0.60 to 1.09) p=0.163 |
| Ages 61–80 | 321 | £451.29 (£336.93) | 1.11 (0.93 to 1.34) p=0.246 | £629.87 (£730.63) | 0.85 (0.61 to 1.18) p=0.336 |
| Ages 81+ | 44 | £442.46 (£146.20) | 1.15 (0.90 to 1.47) p=0.252 | £712.65 (£973.69) | 0.99 (0.64 to 1.53) p=0.976 |

*Exponentiated parameter estimate (SE) from a generalised linear model with log-link and gamma distributed errors including gender, age, ethnicity, employment status, smoking status, units of alcohol per week, body mass index, bilateral CTS, comorbidities and comorbidity score.
CTS, carpal tunnel syndrome; NCS, nerve conduction studies; NHS, National Health Service.

nocturnal night and waking from pain). It should also be noted that pain is not always the predominant symptom.[8]

To the best of our knowledge, this is the first study to explore the association of self-reported and objectively graded symptom severity with psychological distress and quality of life using a multivariable model while adjusting for the effect of other confounders. It is also the first study to measure prospectively the NHS resource use and societal costs of patients with CTS. We included a wide range of variables which could account for increased psychological distress and decreased quality of life including comorbidities weighted by how activity limiting they are, smoking status, alcohol consumption, age, gender, ethnicity, employment status and duration of CTS.

A surprising finding is that objective severity assessed by NCS showed only a weak but negative association with anxiety, and there were no significant associations between NCS and HRQoL or NCS and depression. There has been much debate over the value of NCS in the diagnosis of CTS,[1 22–24] and in the UK not all patients have NCS.[7] Several studies have examined the association of self-reported symptoms and function with objective severity from electrodiagnostic findings. Most have used simple correlations with the exception of Chan et al.[25] The latter used a multivariable model controlling for several demographic variables, as well as psychological measures including depression, somatisation and pain-related catastrophising, in a sample of 215 patients with CTS. They conclude that subjective symptoms

and objective severity are independent of each other. One possible explanation is that NCS can only measure the velocity and amplitude of large myelinated fibres that is sensory and motor axons. Yet CTS also affects the small unmyelinated fibres,[26] and it is these which account for the neuropathic pain component manifesting as pain during daytime and night-time. Only three out of the six questions in the CTS-6 address sensory symptoms (daytime numbness or tingling, night-time numbness or tingling, and waking from numbness and tingling), with the other three questions relating to pain at night, in the day and waking due to pain. Patients with more severe electrodiagnostic CTS (grades 5 and 6) have numbness and even muscle wasting but often do not have any 'positive' symptoms such as tingling or pain resulting in a lower overall self-reported symptom score. It has also been suggested that those with more severe compression often have less subjective symptoms, especially in older patients in whom there is a natural age-related decline in sensory function and reduced pain sensitivity[27] but greater functional disability.[28] This may also explain the significant association observed between NCS grade and hand function (MHQ) as severe numbness and thenar wasting found in grades 5 and 6 are likely to have a greater impact on hand function.

Another reason for the lack of association with nerve conduction results is that CTS causes small fibre loss not detected by NCS and, surprisingly, in some people with more severe nerve entrapment subjective symptoms and function improve suggesting an adaptive mechanism.[26]

The slight association of anxiety with very mild or mild electrophysiological abnormalities of the median nerve CTS could represent disproportionate concern about mild symptoms from mild disease (the essence of anxiety), misdiagnosis of CTS with incidental mild electrophysiological abnormalities or a reaction to dismissive clinicians.

Our analysis, being cross sectional, cannot address the question of whether anxiety and depression are the cause or the effect of self-reported symptom severity. There may well be a bidirectional relationship between symptom severity and psychological distress, where each contributes to the development and is a consequence of the other. Neither can we conclude from these data that if symptom severity improves after treatment then psychological distress or quality of life would also improve. A possible explanation for the significant independent association between self-reported symptoms, psychological status and HRQoL is that people either consistently underplay or exaggerate their symptoms, irrespective of whether these are physical, psychological or global HRQoL. This may also explain the increased costs, both from an NHS and personal perspective, driven by increased treatment-seeking behaviour in those who perceive their symptoms as worse.

HRQoL was markedly lower in those with very severe self-reported symptoms (scores 4–5 on CTS-6). Their mean health utility of 0.43 is much lower than that reported in other studies of patients with CTS prior to surgery ranging from 0.74 to 0.81 using the EQ-5D and 0.69 using the SF-36 (Short Form-36) or SF-6D (Short Form-6 dimension).[29–31] In fact, the mean utility index for this group is closer to that reported by EQ-5D in people with serious diseases like severe heart failure,[32] moderate to severe psychotic illness[33] and digestive system cancers.[34]

## Strengths and limitations

The strengths of the present study include its prospective design, multicentre recruitment and a large sample size (>700) in whom a wide range of prespecified sociodemographic and clinical variables were collected at baseline. Using a multivariable model mitigates the potentially confounding factors such as age, comorbidities, and so on, by adjusting for these and thus testing the independence of any associations between self-reported or neurophysiological severity and psychological distress, hand function or quality of life.

There are also limitations, namely that the data collection was reliant on patient self-report which is open to recall bias, and data quality could not be independently verified. For example, duration of symptoms may be difficult to attribute separately to each hand in the case of bilateral symptoms or to remember accurately over a longer period. Patients were recruited through secondary care having been referred by a primary care physician. It is therefore possible that our study sample is biased towards those actively seeking treatment for their CTS. Long-term follow-up over 2 years is under way and will allow us to explore predictors of treatment-related outcomes and cost. We were also not able to explain why costs were higher for the lowest and highest CTS-6 grades, although it should be borne in mind that these costs relate only to the 3 months prior to entry in the study. The 2-year follow-up data will provide a more robust analysis. Finally, our sample is biased towards an older age group and may not be representative of the typical clinical population with CTS.

## CONCLUSIONS

This study shows that even when accounting for potential confounders such as, for example, comorbidities or age, a strong and highly significant association remains between symptom severity and psychological status and HRQoL. This means that those with more severe symptoms are also more likely to present with worst mental health, poorer hand function and lower quality of life, thus placing a significant burden on the individual, health services and society. Therefore, access to timely and effective diagnostics and effective treatment before symptoms become severe or very severe is paramount. NCS remains a useful tool to assess disease severity, which in turn should be used to determine whether surgery is indicated or not; however, NCS should not be used as a measure of the impact of CTS on the person. Patients presenting with only mild disease severity on NCS but who exhibit anxiety or depression may need to be offered additional psychological support.

**Author affiliations**
[1]Faculty of Medicine and Health Sciences, School of Health, Sciences, University of East Anglia, Norwich, UK

[2]Department of Clinical Neurophysiology, Norfolk and Norwich University Hospitals NHS Foundation Trust, Norwich, UK
[3]Department of Clinical Neurophysiology, MRC Centre for Neuromuscular Diseases, London, UK
[4]Cambridge Centre for Health Services Research, Institute of Public Health, University of Cambridge, Cambridge, UK
[5]Faculty of Medicine and Health Sciences, Norwich Medical School, University of East Anglia, Norwich, UK

**Acknowledgements** We thank the patients who participated in this study; the local Principal Investigators at the following collaborating NHS sites: Julian Blake and Adrian Tearle (Norfolk and Norwich University Hospitals NHS Trust), Ann Harvey (Ipswich Hospital), Stephen Scott (James Paget Hospital, GtYarmouth), Jeremy Bland (Kings College Hospital), Lionel Christopher Bainbridge (Royal Derby Hospital); lay members from the local Patient and Public Involvement in Research group: Jenny Griffiths and Sue Spooner; Tony Dyer and Antony Colles for database support (Norwich Clinical Trials Unit).

**Contributors** CJH conceived the PALMS cohort study and obtained funding. CJH, JB, LS and ECFW developed the protocol. JH conducted data collection and data entry. LS, AS and ECFW undertook the statistical analyses. All authors contributed to the drafting of the manuscript and have read and approved the final manuscript.

**Funding** CJH was funded by the National Institute for Health Research (NIHR) through an NIHR Senior Research Fellowship. ECFW was funded by the NIHR Cambridge Biomedical Research Centre. The funders of the study had no role in study design, data collection, data analysis, data interpretation or writing of the report. The views and opinions expressed therein are those of the authors and do not necessarily reflect those of the NIHR, NHS or the Department of Health.

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
