## [Reviewer comments · BMJ Open]

ARTICLE DETAILS

TITLE (PROVISIONAL)	Association of psychological distress, quality of life and costs with carpal tunnel syndrome severity: a cross-sectional analysis of the PALMS cohort
AUTHORS	Jerosch-Herold, Christina; Houghton, Julie; Blake, Julian; Shaikh, Anum; Wilson, Edward; Shepstone, Lee

VERSION 1 – REVIEW

REVIEWER	David Ring Dell Medical School -- The University of Texas at Austin, USA
REVIEW RETURNED	19-May-2017

GENERAL COMMENTS	1. Page 4, lines 15-16: You must distinguish between symptoms/limitations (illness) and disease (median neuropathy at the carpal tunnel). Symptoms might be relieved or wax and wane, but the disease marches on and leads to atrophy, weakness and static numbness over several decades.2. Page 4, lines 15-16: The word “nonoperative” is preferred over “conservative.”3. Page 4, lines 15-16: “Require” is the wrong word to use with respect to symptoms. No one requires surgery for pain for instance. You might indicate that without surgery CTS can progress to permanent nerve damage.4. Page 4, lines 36-37: What type of “severity” is the debate about? The illness or the disease? Symptoms or pathophysiology? I think there is little debate about the indications for CTR in terms of pathophysiology. The only debate there might be that advanced CTS has permanent nerve damage and surgery can be disappointing.5. Page 4, lines 41-47: It’s not clear from the Introduction why you looked at psychological factors? I would argue that patients and surgeons often consider surgery based on symptoms when they should base surgery on pathophysiology. Your confirmatory study that symptoms are related to stress, distress, and less effective coping strategies would further establish that symptom severity is not a good basis for considering surgery in patients with CTS. Given the nature of CTS as a disease where surgery might be considered “necessary” because preserving sensation in the median nerve innervated digits is so important, we can apply the finding that symptom intensity is not a good guide to the role of surgery to other hand conditions—most of which are highly discretionary and preference-sensitive.
---

	6. Page 5, line 3: I recommend you separate symptoms from pathophysiology. Describe the diagnosis of idiopathic median neuropathy at the carpal tunnel using electrodiagnostic testing (objective measurable pathophysiology). Distinguish that from symptoms. Presumably everyone in this cohort had testing for symptoms. So for this study you can disregard symptoms as a part of the diagnosis and just focus on electrodiagnostic objective verification of disease in a cohort of patients with symptoms. 7. Page 5, line 8: CTS is never “due” to hypothyroidism or diabetes. It’s debatable whether it’s even related. Severe hypothyroid is a thing of the past. Omit this. 8. Results: People with more advanced disease were less anxious and more adaptive. 9. Lines 38- There are plenty of studies of symptoms intensity and limitations in people with CTS. Many of them look at electrodiagnostic severity. Here’s two: 23027833, 23890497. Keep looking! 10. I completely disagree with your conclusions. Your study—building on lots and lots of evidence on CTS, hand surgery, and medicine in general—shows that there is a substantial divide between pathophysiology (disease) and symptoms/limitations (illness). In the biopsychosocial paradigm, disease is treated appropriately (e.g. penicillin for strep throat, surgery for moderate CTS on EMG) and symptoms and limitations are managed by addressing stress, distress, and less effective coping strategies. It would be a major step backwards to say that we should treat pathophysiology based on symptoms. That represents a misdiagnosis and will continue the shameful undertreatment of the psychosocial aspects of illness.
--	--

REVIEWER	Cesar Fernandez-de-las-Peñas Universidad Rey Juan Carlos, Spain
REVIEW RETURNED	28-May-2017

GENERAL COMMENTS	This paper covers an interesting topic but it needs further revision and several sections should be completely rewritten. The introduction is poor. Authors should discuss current knowledge on anxiety and depression and the other variables in CTS and chronic pain. The current introduction does not help the readers for the current study. Methods: If more data from the same sample size has been published it should be included in the methods. This section is clearly poorly described. All outcomes should be described and justify their use in CTS. More description of the EMG analysis and classification of the patients is clearly needed. I presume that authors first conducted correlation analysis since this is needed for doing after linear regression models. The statistical analysis section should be clarified and expanded. Results: If almost 80% of the patients had normal HADS scores, depression and anxiety could not be determined, so the results would be not valid. Authors should include r and adjust R values of the linear regression models, and not just the P values.
---

	We cannot say that severity is associated to anxiety and depression where there were not these psychological aspects. Please clarify in the discussion. Authors should include direct association coefficients before linear regression. There is no data on how costs were analyzed. It is important to include a brief comment in the methods, and not just refer to the published protocol. Discussion: More discussion on depression and anxiety is clearly needed. There are some studies including sample sizes of 200 patients (Clin J Pain 2014, Pain Med 2015) investigating the role of depression in function. These should be also included. Authors should include some hypothesis why depression can be related to symptom's severity.
--	--

REVIEWER	Isam Atroshi Lund University, Sweden
REVIEW RETURNED	07-Jun-2017

GENERAL COMMENTS	This is a large study with new interesting information that is useful to clinicians and researchers in the field of carpal tunnel syndrome. The manuscript is well-written and generally clear. Introduction: The statement that surgical decompression rates after initial corticosteroid injection “vary by the country and referral criteria” is surprising because they should depend on efficacy of treatment in resolving patients’ symptoms rather than in which country the patients live. Probably these variations reflect the study design and other study-related factors rather than country. Methods: Was the study registered? Were the primary and secondary analyses for this study prespecified in a study protocol before patients were recruited. What was the rationale of recruiting patients from 4 neurophysiological departments but only 1 hand surgery center? Usually surgeons in clinical practice refer patients when they need confirmation (ie clinical diagnosis not adequately clear to proceed to surgery) while patients with typical history are often treated without nerve conduction tests. If there are data available it would be helpful to report the proportion of patients referred to NCS by primary care physician and by surgeons. Who diagnosed the patients with CTS and who invited them to participate? Who graded the NCS results? Do these 4 neurophysiological units use identical testing methods and identical reference values for what is abnormal? In patients with bilateral CTS, how was the worst hand determined, by the patient or by CTS-6 score, or nerve conduction test results? A description of the HADS scoring would be helpful. Was “time of work due to CTS” self-reported, like number of days? Needs some clarification. It would be helpful to describe the Bland criteria because not all readers are familiar with them and they are important in the analyses and conclusions. Usually the EQ-5D value that is derived from weights is called the EQ-5D “index” not score.
---

Considering that false positive NCS may be more common than believed, is it possible that the paradoxical association between lowest NCS severity (near normal) and high anxiety may be incorrect CTS diagnosis?

Why do NHS and societal costs increase with self-reported CTS severity?

Are younger patients with clinical diagnosis of CTS less likely to have abnormal NCS than older patients? This is important because the severity of self-reported symptoms are likely to be similar or even as sometimes suggested that old persons with CTS may even have less symptoms despite severe median nerve dysfunction. Is it known how large proportion of patients that are not referred to NCS but treated entirely based on clinical diagnosis? are there any data about this in the study region?

Table 1

The mean age of the patients (60 years) is somewhat higher than the average CTS population and only about 50% working, do we have any data about the group that were eligible but did not participate in the study?

Table 2

The authors show data about activity limitations (MHQ) but these are not further addressed. It would be interesting to see for example the relationship between activity limitations and CTS-6 and NCS. Anxiety and depression categories show that only a small proportion have at least moderate (18% for anxiety and 8% for depression).

Table 4

It would be helpful to add number of patients for each of the CTS-6 score and NCS grade categories. It would also be interesting to add the mean CTS-6 score for each NCS grade. 95% confidence interval is usually easier to interpret than SE. Any correlation between age and NCS grade?

Table 5

Is it correctly interpreted that patients with CTS-6 score of 1 (ie have no CTS symptoms according to the CTS-6 scale) have the highest costs? Any explanations for this? It may indicate that these patients have some other diagnosis than CTS.

In both Table 4 and 5 the CTS-6 score categories (for example 1-2 and then 2-3 etc) should be more accurately specified without overlapping.

It is unclear how to interpret the data in Table 5 regarding "Age & CTS-6" and "Age & NCS grade".

The reference list does not follow a uniform style, needs to be checked.

Figure 1

Were 1918 patients eligible? 140 declined and 820 accepted, who were the remaining patients?

Discussion

There are longitudinal CTS studies that have measured mental health aspects, for example using the SF-36, before and after surgery, the authors could discuss the findings from these studies.

VERSION 1 – AUTHOR RESPONSE

Reviewer: 1

Reviewer Name: David Ring

Institution and Country: Dell Medical School -- The University of Texas at Austin, USA
Competing Interests: None.

Comment 1. Page 4, lines 15-16: You must distinguish between symptoms/limitations (illness) and disease (median neuropathy at the carpal tunnel). Symptoms might be relieved or wax and wane, but the disease marches on and leads to atrophy, weakness and static numbness over several decades.

authors response: Thank you we agree and have changed the sentence to : Whilst in some people symptoms of CTS resolve spontaneously or respond to non-operative treatment by corticosteroid injection, many undergo surgical decompression to prevent irreversible nerve damage.

Comment 2. Page 4, lines 15-16: The word “nonoperative” is preferred over “conservative.”

authors response: Changed to ‘non-operative’.

Comment 3. Page 4, lines 15-16: “Require” is the wrong word to use with respect to symptoms. No one requires surgery for pain for instance. You might indicate that without surgery CTS can progress to permanent nerve damage.

authors response: Changed as suggested.

Comment 4. Page 4, lines 36-37: What type of “severity” is the debate about? The illness or the disease? Symptoms or pathophysiology? I think there is little debate about the indications for CTR in terms of pathophysiology. The only debate there might be that advanced CTS has permanent nerve damage and surgery can be disappointing.

authors response: Thank you this indeed an important distinction and it is the lack of clarity in the existing guidelines about what is meant by severity which compounds the problem. Some define it using neurophysiology, others rely on patient-reported severity. We have therefore placed ‘moderate to severe’ in quote marks as we are citing from these documents and added a sentence highlighting this inconsistency in the terms used.

Comment 5. Page 4, lines 41-47: It’s not clear from the Introduction why you looked at psychological factors? I would argue that patients and surgeons often consider surgery based on symptoms when they should base surgery on pathophysiology. Your confirmatory study that symptoms are related to stress, distress, and less effective coping strategies would further establish that symptom severity is not a good basis for considering surgery in patients with CTS. Given the nature of CTS as a disease where surgery might be considered “necessary” because preserving sensation in the median nerve innervated digits is so important, we can apply the finding that symptom intensity is not a good guide to the role of surgery to other hand conditions—most of which are highly discretionary and preference-sensitive.

authors response: This is also commented on by reviewer 2 and we have now added further justification in the introduction of why we have examined psychological factors and HRQoL. We also agree that symptoms are not a good indicator for how to treat CTS but would maintain that they are a measure of disease impact and therefore can inform ‘whether’ to treat someone.

Comment 6. Page 5, line 3: I recommend you separate symptoms from pathophysiology. Describe the diagnosis of idiopathic median neuropathy at the carpal tunnel using electrodiagnostic testing (objective measurable pathophysiology). Distinguish that from symptoms. Presumably everyone in this cohort had testing for symptoms. So for this study you can disregard symptoms as a part of the diagnosis and just focus on electrodiagnostic objective verification of disease in a cohort of patients with symptoms.

authors response: All patients were examined for signs and symptoms and objective nerve conduction studies. It is common practice that neurophysiologists offer an interpretation of the NCS results alongside signs and symptoms in their written reports to the referrer. We have reworded this sentence for clarity.

Comment 7. Page 5, line 8: CTS is never “due” to hypothyroidism or diabetes. It’s debatable whether it’s even related. Severe hypothyroid is a thing of the past. Omit this.

authors response: These were the prespecified inclusion criteria in the published protocol and we feel it is important to report these transparently. However we have changed the sentence so that the attribution of CTS to these conditions is removed.

Comment 8. Results: People with more advanced disease were less anxious and more adaptive.

authors response: Yes, we agree that they were less anxious however we cannot say that they were more adaptive as we did not measure that.

Comment 9. Lines 38- There are plenty of studies of symptoms intensity and limitations in people with CTS. Many of them look at electrodiagnostic severity. Here’s two: 23027833, 23890497. Keep looking!

authors response: We assume the reviewer is referring to the discussion section line 38 and that the claimed ‘primacy’ is being questioned here. We have reworded this section now and included further studies as suggested and discussed our findings in relation to these.

Comment 10. I completely disagree with your conclusions. Your study—building on lots and lots of evidence on CTS, hand surgery, and medicine in general—shows that there is a substantial divide between pathophysiology (disease) and symptoms/limitations (illness). In the biopsychosocial paradigm, disease is treated appropriately (e.g. penicillin for strep throat, surgery for moderate CTS on EMG) and symptoms and limitations are managed by addressing stress, distress, and less effective coping strategies. It would be a major step backwards to say that we should treat pathophysiology based on symptoms. That represents a misdiagnosis and will continue the shameful undertreatment of the psychosocial aspects of illness.

authors response: We agree with the reviewer’s stance regarding the approach to medical treatment of CTS and are unclear how our conclusions contradict this. We don’t believe that we have said that pathophysiology should be treated based on symptoms. We have reworded the conclusion to clarify this distinction between whether someone needs treatment for CTS and what treatment should be given and the likely prognosis. The former should be informed by symptoms and impact on function, whereas the latter should be informed by pathophysiology e.g. if grade 1 then non-operative treatment may be tried.

Reviewer: 2

Reviewer Name: Cesar Fernandez-de-las-Peñas Institution and Country: Universidad Rey Juan Carlos, Spain Competing Interests: None declared

Comment: This paper covers an interesting topic but it needs further revision and several sections should be completely rewritten.

The introduction is poor. Authors should discuss current knowledge on anxiety and depression and the other variables in CTS and chronic pain. The current introduction does not help the readers for the current study.

authors response: We have added a section including previous research on CTS and psychological aspects as well as clarified the rationale for our analysis. We also want to point out that the focus of our study was not chronic pain and whilst pain features in the measurement of symptoms in the CTS-6 we cannot say that it is 'chronic'.

Methods

Comment: If more data from the same sample size has been published it should be included in the methods. This section is clearly poorly described. All outcomes should be described and justify their use in CTS. More description of the EMG analysis and classification of the patients is clearly needed. I presume that authors first conducted correlation analysis since this is needed for doing after linear regression models. The statistical analysis section should be clarified and expanded.

authors response: No other data from this sample has been published. The study is still in follow-up and analysis of that data is still to be completed.

The justification for the included outcome measures is fully described in the previously published protocol which has been referenced.

We did not undertake EMG analysis but nerve conduction studies (NCS), however we have given further details on the methods for classifying objective severity by NCS in a supplementary table (also suggested by reviewer 3).

We did examine bi-variate correlations between CTS measures and our selected outcomes. These ranged in magnitude from 0.03 to 0.56. However, the statistical analysis was 'hypothesis driven', i.e. we wished to examine the strength of relationship between CTS measures and outcomes, rather than 'exploratory', i.e. assessing which measures were related to outcomes. As such, there was no 'model construction' or process of selection of explanatory variables; we included CTS-6 and NCS measures as predictors and then further variables as potential confounders. We do not feel that the bi-variate correlations contribute substantially to the paper and have not included them.

Results

Comment: If almost 80% of the patients had normal HADS scores, depression and anxiety could not be determined, so the results would be not valid. Authors should include r and adjusted R values of the linear regression models, and not just the P values. We cannot say that severity is associated to anxiety and depression where there were not these psychological aspects. Please clarify in the discussion. Authors should include direct association coefficients before linear regression.

authors response: It is correct that nearly 80% of participants had normal HADS depression scores. However, this information was added to provide interpretation of the scores. The modelling was not done with the categories as outcomes but rather with respect to the scores themselves. For both the HADS anxiety and HADS depression scores, the participant scores ranged from 0 to 20.

R -squared values have now been included in table 4. As indicated above, we have not included bi-variate correlation co-efficients.

There is no data on how costs were analyzed. It is important to include a brief comment in the methods, and not just refer to the published protocol.

authors response: Methods for calculating costs have been added under the statistical analysis section

Discussion

Comment: More discussion on depression and anxiety is clearly needed. There are some studies including sample sizes of 200 patients (Clin J Pain 2014, Pain Med 2015) investigating the role of depression in function. These should be also included. Authors should include some hypothesis why depression can be related to symptom's severity.

authors response: We have included the suggested references in the discussion. We had already given some plausible explanations for the possible relationship between depression and symptom severity including this possibly being a bidirectional relationship.

Reviewer: 3

Reviewer Name: Isam Atroshi

Institution and Country: Lund University, Sweden Competing Interests: 'None declared'

This is a large study with new interesting information that is useful to clinicians and researchers in the field of carpal tunnel syndrome.

The manuscript is well-written and generally clear. authors response: Thank you

Introduction

Comment: The statement that surgical decompression rates after initial corticosteroid injection "vary by the country and referral criteria" is surprising because they should depend on efficacy of treatment in resolving patients' symptoms rather than in which country the patients live. Probably these variations reflect the study design and other study-related factors rather than country.

authors response: Yes we agree that these should be the same however there are differences in access to surgical decompression in the UK National Health Service depending on the specific local clinical commissioning criteria. We have reworded this sentence as suggested.

Methods

Comment: Was the study registered? Were the primary and secondary analyses for this study prespecified in a study protocol before patients were recruited.

authors response: The PALMS cohort study was not registered but the protocol has been published (referenced in text). The primary analysis, which is to develop multivariable models for predictors of outcome and predictors of cost are pre-specified. The focus of this paper is the cross-sectional analysis which is a secondary analysis of the baseline data for the cohort.

Comment: What was the rationale of recruiting patients from 4 neurophysiological departments but only 1 hand surgery center? Usually surgeons in clinical practice refer patients when they need confirmation (ie clinical diagnosis not adequately clear to proceed to surgery) while patients with typical history are often treated without nerve conduction tests. If there are data available it would be helpful to report the proportion of patients referred to NCS by primary care physician and by surgeons.

authors response: We chose participating centres where neurophysiology testing is undertaken in all patients referred from primary care with a suspected diagnosis of CTS as this was considered the best time point at which to enrol patients into this prospective study. We have reworded that sentence to clarify that these were secondary care sites and that patients were referred from primary care.

Comment: Who diagnosed the patients with CTS and who invited them to participate? Who graded the NCS results?

authors response: Either a neurophysiologist or hand surgeon made the diagnosis based on the combination of signs and symptoms and NCS reports. The grading of all NCS reports according to the Bland criteria was done by the first author (CJH).

Comment: Do these 4 neurophysiological units use identical testing methods and identical reference values for what is abnormal?

authors response: The method and equipment for orthodromic testing is the same in all units and standard operating procedures were followed for identifying eligible patients who were at least grade 1 based on the Bland's criteria (now given in a supplementary table).

Comment: In patients with bilateral CTS, how was the worst hand determined, by the patient or by CTS-6 score, or nerve conduction test results?

authors response: Worst hand was determined by the patient based on CTS-6 score and we have added this in the text.

Comment: A description of the HADS scoring would be helpful.

authors response: We have added this to text under methods/data collection.

Comment: Was "time of work due to CTS" self-reported, like number of days? Needs some clarification.

authors response: Yes it is patient reported and in days – we have amended this.

Comment: It would be helpful to describe the Bland criteria because not all readers are familiar with them and they are important in the analyses and conclusions.

authors response: We have provided this as a supplementary file/table with the criteria (also suggested by reviewer 2).

Comment: Usually the EQ-5D value that is derived from weights is called the EQ-5D "index" not score.

authors response: Thank you for pointing this out, we have amended as suggested.

Comment: Considering that false positive NCS may be more common than believed, is it possible that the paradoxical association between lowest NCS severity (near normal) and high anxiety may be incorrect CTS diagnosis?

authors response: Thank you, yes that is also a possible explanation and we have included this in the discussion.

Comment: Why do NHS and societal costs increase with self-reported CTS severity?

authors response: We have added a sentence to elaborate on this within the results and believe that in the discussion we did give a possible explanation as follows: 'This may also explain the increased costs, both from an NHS and personal perspective, driven by increased treatment-seeking behaviour in those who perceive their symptoms as worse.'

Comment: Are younger patients with clinical diagnosis of CTS less likely to have abnormal NCS than older patients? This is important because the severity of self-reported symptoms are likely to be similar or even as sometimes suggested that old persons with CTS may even have less symptoms despite severe median nerve dysfunction.

authors response: We did consider whether the more severe NCS grades were from older patients who, as you say, sometimes report less severe symptoms, however we found that the distribution of grades 5 and 6 were across the age span. Also the multivariable model accounts for several variables including age and so any association between NCS and psychological status or QoL is independent of age.

Comment: Is it known how large proportion of patients that are not referred to NCS but treated entirely based on clinical diagnosis? are there any data about this in the study region?

authors response: Indeed, it is the case that many patient undergo non-operative and operative treatment without ever having had any NCS. We do not have data on this for our cohort as we deliberately selected centres, where patients could be recruited immediately after having CTS confirmed with NCS.

Comment: Table 1

The mean age of the patients (60 years) is somewhat higher than the average CTS population and only about 50% working, do we have any data about the group that were eligible but did not participate in the study?

authors response: Yes we concur this is not typical but because this was a research study as opposed to an analysis of routinely collected clinical data we depended on patients who gave consent to be enrolled in this study. This has biased the sample more towards those retired and older. We do not have any information on those who did not respond as we do not have ethical approval to access their clinical records. We have added a comment in the discussion acknowledging this limitation.

Comment: Table 2

The authors show data about activity limitations (MHQ) but these are not further addressed. It would be interesting to see for example the relationship between activity limitations and CTS-6 and NCS.

authors response: Yes we agree and have now extended the statistical modelling to include the MHQ as outcomes. These results are now shown in table 4(b), with the original table 4 now being re-labelled table 4(a).

Anxiety and depression categories show that only a small proportion have at least moderate (18% for anxiety and 8% for depression).

authors response: Yes, this is correct.

Comment: Table 4

It would be helpful to add number of patients for each of the CTS-6 score and NCS grade categories.

authors response: These numbers have now been added to tables 4 and 5. We would like to note that when doing this for tables 4 and 5 we found some minor discrepancies and the analysis of costs had to be re-run. As a result the values in the text and table 5 changed slightly however do not affect the overall results or interpretation.

Comment: It would also be interesting to add the mean CTS-6 score for each NCS grade.

authors response: The mean (and standard deviation) CTS-6 score by NCS grade is as follows:

Grade Mean (SD)

1 2.69 (0.86)

2 2.79 (0.85)

3 2.87 (0.91)

4 3.00 (0.79)

5 3.06 (0.79)

6 2.89 (0.78)

These have now been added to the text.

Comment: 95% confidence interval is usually easier to interpret than SE.

authors response: We agree and have now included 95% confidence intervals alongside the parameter estimates in table 4 and 5.

Comment: Any correlation between age and NCS grade?

authors response: There was a weak correlation between age and NCS grade ($r = 0.367$, $p < 0.001$). We are unsure if this adds to the manuscript and have not included it within the results.

Comment: Table 5

Is it correctly interpreted that patients with CTS-6 score of 1 (ie have no CTS symptoms according to the CTS-6 scale) have the highest costs? Any explanations for this? It may indicate that these patients have some other diagnosis than CTS.

authors response: This is addressed in the discussion section under 'Strengths and limitations'.

Comment: In both Table 4 and 5 the CTS-6 score categories (for example 1-2 and then 2-3 etc) should be more accurately specified without overlapping.

authors response: This is a good point and we have now modified the tables to clarify this.

Comment: It is unclear how to interpret the data in Table 5 regarding "Age & CTS-6" and "Age & NCS grade".

authors response: We have added this to the results section.

Comment: The reference list does not follow a uniform style, needs to be checked.

authors response: We have checked and corrected all references

Comment: Figure 1

Were 1918 patients eligible? 140 declined and 820 accepted, who were the remaining patients?

authors response: To clarify this we have amended this figure to include information regarding the remaining patients who did not respond.

Discussion

Comment:

There are longitudinal CTS studies that have measured mental health aspects, for example using the SF-36, before and after surgery, the authors could discuss the findings from these studies.

authors response: We did provide some comparative data on health utility using the EQ-5D but have added a further reference using the SF-36 as suggested.

Reviewer: 4

Reviewer Name: Luca Padua

Institution and Country: Università Cattolica, Fondazione Don Gnocchi, Italy Competing Interests: Neurophysiology, peripheral nervous system, ultrasound, neuro-rehabilitation

Comment: Very good work! Needed and adequately conducted.

authors response: Thank you very much

VERSION 2 – REVIEW

REVIEWER	David Ring Dell Medical School -- The University of Texas at Austin, USA
REVIEW RETURNED	14-Jul-2017

GENERAL COMMENTS	1. Intro, first paragraph: This background information is not particularly necessary or helpful. Some of it introduces debatable concepts such as the idea that idiopathic median neuropathy at the carpal tunnel can resolve spontaneously or be cured with a steroid injection. This represents confusion of the illness (symptoms and limitation) with the disease (the pathophysiology of the median nerve). As your data clearly shows symptoms correlate with psychosocial factors, not pathophysiology. That means that people with pathophysiology may not experience symptoms. With a structural disease like carpal tunnel syndrome, symptoms may resolve or become less bothersome, but the disease most likely persists. You may do people a disservice by suggesting that median neuropathy can resolve on it's own. Such a person might adapt and then end up with advanced neuropathy that is permanent. I realize there is room for debate here. So my advice is to just delete this background information. You can do without it. 2. Introduction, paragraph 2. Moderate to severe refers to electrodiagnostic testing (pathophysiology). Since symptoms don't correlate with disease severity, greater symptoms indicate opportunities for relief of distress and work on more effective coping strategies. Greater symptoms should not affect the indication for disease-modifying surgery.
---

	3. I disagree with your conclusions. NCS should be used to assess disease severity and disease severity should be used to determine the indication for surgery. Greater symptoms and limitations than expected for a given disease severity should prompt a screen for psychological distress, greater social stressors, and less effective coping strategies. Treatment can then be tailored to address these aspects of the illness. Surgery for normal or mild NCS would be likely to result in a missed opportunity to treat depression or catastrophic thinking. Avoiding surgery because symptoms aren't severe enough could lead to numbness, weakness, and atrophy that could have been avoided. At a minimum, this possibility should be acknowledged in your paper, even it does not fit your bias.
--	---

REVIEWER	Cesar Fernandez-de-las-Peñas Universidad Rey Juan Carlos, Spain
REVIEW RETURNED	14-Jul-2017

GENERAL COMMENTS	Authors have edited all comments properly
---

REVIEWER	Isam Atroshi Lund University, Sweden
REVIEW RETURNED	31-Jul-2017

GENERAL COMMENTS	The questions have been addressed very well. Two minor issues: 1. Because the CTS-6 scale ranges from 1 to 5, the mean CTS-6 score can never be less than 1.0 and therefore the score category "0 – 1.00" should be corrected to "1.00" in all tables. It may be of interest when discussing the results for this category to highlight that a mean score of 1 means all patients in that category had no symptoms of CTS as measured by the CTS-6 (numbness, tingling and pain experienced during the past 2 weeks). 2. In the Discussion about health utility measures, a statement mentions the exact value from a previous study and refers to it as the SF-36 but it is more accurately the SF-6D index value (even though the SF-6D is derived from the SF-36).
---

VERSION 2 – AUTHOR RESPONSE

Reviewer: 1

Reviewer Name: David Ring

Institution and Country: Dell Medical School -- The University of Texas at Austin, USA
Competing Interests: None.

Comment 1. Intro, first paragraph: This background information is not particularly necessary or helpful. Some of it introduces debatable concepts such as the idea that idiopathic median neuropathy at the carpal tunnel can resolve spontaneously or be cured with a steroid injection. This represents confusion of the illness (symptoms and limitation) with the disease (the pathophysiology of the median nerve). As your data clearly shows symptoms correlate with psychosocial factors, not pathophysiology. That means that people with pathophysiology may not experience symptoms. With a structural disease like carpal tunnel syndrome, symptoms may resolve or become less bothersome, but the disease most likely persists. You may do people a disservice by suggesting that median neuropathy can resolve on it's own. Such a person might adapt and then end up with advanced

neuropathy that is permanent. I realize there is room for debate here. So my advice is to just delete this background information. You can do without it.

Authors response: We have removed any reference to CTS resolving by its own or in response to steroid injections, however we feel that it would not be appropriate to remove this whole first paragraph as it sets an important context for the next paragraph and the study.

Comment 2. Introduction, paragraph 2. Moderate to severe refers to electrodiagnostic testing (pathophysiology). Since symptoms don't correlate with disease severity, greater symptoms indicate opportunities for relief of distress and work on more effective coping strategies. Greater symptoms should not affect the indication for disease-modifying surgery.

Authors response: the words 'moderate to severe' have been used here in the context of how the clinical commissioning group guidelines use them and have been placed in quotation marks. We are unclear what else is being requested here.

Comment 3. I disagree with your conclusions. NCS should be used to assess disease severity and disease severity should be used to determine the indication for surgery. Greater symptoms and limitations than expected for a given disease severity should prompt a screen for psychological distress, greater social stressors, and less effective coping strategies. Treatment can then be tailored to address these aspects of the illness. Surgery for normal or mild NCS would be likely to result in a missed opportunity to treat depression or catastrophic thinking. Avoiding surgery because symptoms aren't severe enough could lead to numbness, weakness, and atrophy that could have been avoided. At a minimum, this possibility should be acknowledged in your paper, even it does not fit your bias.

Authors' response: we have reworded the sentence about the role of NCS as: ' NCS remains a useful tool in the diagnosis of CTS, and to assess disease severity, which in turn should be used to determine whether surgery is indicated or not'. By using the term 'disease impact' we do not mean 'disease severity', however have changed this to 'impact on the person' to avoid any confusion. We agree that normal and mild NCS grades should not be referred for surgery and added that those with normal or mild NCS exhibiting anxiety or depression may require additional psychological support.

Reviewer: 2

Reviewer Name: Cesar Fernandez-de-las-Peñas Institution and Country: Universidad Rey Juan Carlos, Spain Competing Interests: None Declared
Authors have edited all comments properly

Reviewer: 3

Reviewer Name: Isam Atroshi
Institution and Country: Lund University, Sweden Competing Interests: None declared
The questions have been addressed very well. Two minor issues:

1. Because the CTS-6 scale ranges from 1 to 5, the mean CTS-6 score can never be less than 1.0 and therefore the score category "0 – 1.00" should be corrected to "1.00" in all tables. It may be of interest when discussing the results for this category to highlight that a mean score of 1 means all patients in that category had no symptoms of CTS as measured by the CTS-6 (numbness, tingling and pain experienced during the past 2 weeks).

Authors' response: thank you we agree and have changed 0-1.00 to 1.00 in all the tables. We have also added in the methods/outcome measures section an explanatory sentence that a mean of 1.00 indicates no symptoms.

2. In the Discussion about health utility measures, a statement mentions the exact value from a previous study and refers to it as the SF-36 but it is more accurately the SF-6D index value (even though the SF-6D is derived from the SF-36).

Authors' response: Thank you for pointing this out we have corrected to SF-6D as indicated in one of the references.

We hope that we have addressed all concerns satisfactorily and look forward to hearing from you in due course.

VERSION 3 – REVIEW

REVIEWER	David Ring Dell Medical School--The University of Texas at Austin; The United States of America None. There is no place to have a discussion with the editor, so I will have it here. If the authors can't balance their views with mine (the best option), then you can see from my review that an editorial should accompany this paper.
REVIEW RETURNED	15-Aug-2017

GENERAL COMMENTS	1. Abstract, conclusion: I would argue that directionality is irrelevant and that common sense tells us the relationship is bidirectional. Therefore, Instead of "Further research is needed to understand the possible direction of cause and effect and whether treatment can modify this association", why not say, "Future research can address the impact of approaches and treatments that address psychosocial stressors as well as biomedical factors on relief of symptoms from carpal tunnel syndrome." Your bias is clearly in favor of injections and surgeries that needs to be balanced out to reflect what many of us will read in your data. 2. Discussion, first paragraph. I find it strange the way you present your results. For instance, shouldn't the first sentence say: "We found that at the point of referral for NCS and prior to any treatment patients, greater symptom severity was associated with greater psychological distress and not with worse electrodiagnostic abnormalities." To me this is just additional evidence that resiliency is more important than pathophysiology to good health (fewer symptoms and limitations). 3. Page 10, lower middle. Instead of: "One possible explanation is that NCS can only measure the velocity and amplitude of large myelinated fibres that is sensory and motor axons. Yet CTS also affects the small unmyelinated fibres and it is these which account for the neuropathic pain component manifesting as pain during day and night time", I suggest: "Some people explain the lack of correlation between pathophysiology and symptoms in a biomedical way: that symptoms can be from small unmyelinated fibres that may not be registered as strongly on electrodiagnostic tests." I think you should then balance this theory with it's counter-argument: "But if this were so it should be consistent throughout the range pathophysiology and patients with worse neuropathy should have worse symptoms at each stage."
--

4. Page 11, top: Where you say: "Another reason for the lack of association with nerve conduction results is that CTS causes small fibre loss not detected by NCS and, surprisingly, in some people with more severe nerve entrapment subjective symptoms and function improve suggesting an adaptive mechanism. Although only speculative, it is possible that those with very mild or mild CTS have higher anxiety because they have been misdiagnosed. Another plausible explanation is that in this sample recruited through secondary care, those with very mild CTS represent a more anxious patient group that is also more persistent in seeking medical treatment and referral." I suggest eliminating the redundancy at the start of the paragraph and simply stating: "The slight association of anxiety with very mild or mild electrophysiological abnormalities of the median nerve CTS could represent disproportionate concern about mild symptoms from mild disease (the essence of anxiety); misdiagnosis of carpal tunnel syndrome with incidental mild electrophysiological abnormalities; the speculative disproportionate influence of less detectable small, less myelinated fibers; or a reaction to dismissive clinicians." For me the order of probability goes strongly from start to finish of this sentence. You can order them any way you'd like, but all these possibilities should be presented.

5. Page 11, second paragraph. I find nothing useful in this paragraph. It can be safely omitted.

6. Page 11, paragraph just before "strengths and limitations": There is extensive evidence that PROMs (or HRQOL) are strongly correlated with stress, distress (anxiety/depression), and less effective coping strategies. You should mention this evidence. Most scientists that observe such findings are impressed with the ability of effective coping strategies and limited stress and distress to ameliorate the symptoms and limitations of a given disease. You seem to have an opposite view.

7. My suggestion for Conclusions is as follows:
"This study shows that symptoms and limitations from carpal tunnel syndrome do not correlate with objectively measurable neuropathy, older age, or comorbidities. In other words illness is determined more by psychosocial factors than by pathophysiology. This raises important opportunities for more comprehensive treatment of common hand disorders like carpal tunnel syndrome by adding approaches that limit stress and distress and foster the most effective coping strategies to the typical complement of biomedical treatments such as splints, shots, and surgery."

VERSION 3 – AUTHOR RESPONSE

We thank the reviewer for their comments which we have addressed as follows:

1. Abstract, conclusion: I would argue that directionality is irrelevant and that common sense tells us the relationship is bidirectional. Therefore, Instead of “Further research is needed to understand the possible direction of cause and effect and whether treatment can modify this association”, why not say, “Future research can address the impact of approaches and treatments that address psychosocial stressors as well as biomedical factors on relief of symptoms from carpal tunnel syndrome.” Your bias is clearly in favor of injections and surgeries that needs to be balanced out to reflect what many of us will read in your data.

Authors’ response: we have rephrased this as: Future research is needed to understand the impact of approaches and treatments that address psychosocial stressors as well as biomedical factors on relief of symptoms from carpal tunnel syndrome.

2. Discussion, first paragraph. I find it strange the way you present your results. For instance, shouldn’t the first sentence say: “We found that at the point of referral for NCS and prior to any treatment patients, greater symptom severity was associated with greater psychological distress and not with worse electrodiagnostic abnormalities.” To me this is just additional evidence that resiliency is more important than pathophysiology to good health (fewer symptoms and limitations).

Authors’ response: The proposed sentence excludes hand function and QoL so we have rephrased as suggested included the significant association with quality of life and hand function: We found that at the point of referral for NCS and prior to any treatment, greater symptom severity was associated with greater psychological distress, poorer hand function and quality of life but not with worse electrodiagnostic abnormalities.

3. Page 10, lower middle. Instead of: “One possible explanation is that NCS can only measure the velocity and amplitude of large myelinated fibres that is sensory and motor axons. Yet CTS also affects the small unmyelinated fibres and it is these which account for the neuropathic pain component manifesting as pain during day and night time”, I suggest: “Some people explain the lack of correlation between pathophysiology and symptoms in a biomedical way: that symptoms can be from small unmyelinated fibres that may not be registered as strongly on electrodiagnostic tests.” I think you should then balance this theory with it’s counter-argument: “But if this were so it should be consistent throughout the range pathophysiology and patients with worse neuropathy should have worse symptoms at each stage.”

Authors’ response: we acknowledge that the relationship between NCS and any patient-reported measures is a complex one however the proposed counter-argument regarding proportionality is not necessarily valid and therefore we have not included it.

4. Page 11, top: Where you say: “Another reason for the lack of association with nerve conduction results is that CTS causes small fibre loss not detected by NCS and, surprisingly, in some people with more severe nerve entrapment subjective symptoms and function improve suggesting an adaptive mechanism. Although only speculative, it is possible that those with very mild or mild CTS have higher anxiety because they have been misdiagnosed. Another plausible explanation is that in this sample recruited through secondary care, those with very mild CTS represent a more anxious patient group that is also more persistent in seeking medical treatment and referral.” I suggest eliminating the redundancy at the start of the paragraph and simply stating: “The slight association of anxiety with very mild or mild electrophysiological abnormalities of the median nerve CTS could represent disproportionate concern about mild symptoms from mild disease (the essence of anxiety);

misdiagnosis of carpal tunnel syndrome with incidental mild electrophysiological abnormalities; the speculative disproportionate influence of less detectable small, less myelinated fibers; or a reaction to dismissive clinicians.” For me the order of probability goes strongly from start to finish of this sentence. You can order them any way you’d like, but all these possibilities should be presented.

Authors’ response: The point about an adaptive mechanism is fairly new and therefore we would like to retain this but have added the suggested sentence as follows to explain the higher anxiety in lower disease severity: The slight association of anxiety with very mild or mild electrophysiological abnormalities of the median nerve CTS could represent disproportionate concern about mild symptoms from mild disease (the essence of anxiety), misdiagnosis of carpal tunnel syndrome with incidental mild electrophysiological abnormalities or a reaction to dismissive clinicians

5. Page 11, second paragraph. I find nothing useful in this paragraph. It can be safely omitted.

Authors’ response: We have removed the paragraph as suggested.

6. Page 11, paragraph just before “strengths and limitations”: There is extensive evidence that PROMs (or HRQoL) are strongly correlated with stress, distress (anxiety/depression), and less effective coping strategies. You should mention this evidence. Most scientists that observe such findings are impressed with the ability of effective coping strategies and limited stress and distress to ameliorate the symptoms and limitations of a given disease. You seem to have an opposite view.

Authors’ response: We have addressed this suggestion in part with the sentence on page 11: A possible explanation for the significant independent association between self-reported symptoms, psychological status and HRQoL is that people either consistently underplay or exaggerate their symptoms, irrespective of whether these are physical, psychological or global HRQoL. As we have stated before we did not measure coping strategies or self-efficacy and therefore think the suggestion would go beyond what we can legitimately conclude from the data.

7. My suggestion for Conclusions is as follows:

“This study shows that symptoms and limitations from carpal tunnel syndrome do not correlate with objectively measurable neuropathy, older age, or comorbidities. In other words illness is determined more by psychosocial factors than by pathophysiology. This raises important opportunities for more comprehensive treatment of common hand disorders like carpal tunnel syndrome by adding approaches that limit stress and distress and foster the most effective coping strategies to the typical complement of biomedical treatments such as splints, shots, and surgery.”

Authors’ response: The suggested first sentence cannot be stated with factual accuracy as we did not look at the association between measurable neuropathy and age or comorbidity directly. Further, it is not possible to state that illness is determined more by psychosocial factors than by pathophysiology. We acknowledge in our conclusion that psychological support may well be a useful addition to the traditional biomedical interventions.